

# Effect of storage conditions on SARS-CoV-2 RNA quantification in wastewater solids

Adrian Simpson[1], Aaron Topol[1], Bradley J. White[1], Marlene K. Wolfe[2], Krista R. Wigginton[3] and Alexandria B. Boehm[2]

[1] Verily Life Sciences, South San Francisco, CA, United States of America
[2] Stanford University, Stanford, CA, United States of America
[3] University of Michigan - Ann Arbor, Ann Arbor, MI, United States of America

## ABSTRACT

SARS-CoV-2 RNA in wastewater settled solids is associated with COVID-19 incidence in sewersheds and therefore, there is a strong interest in using these measurements to augment traditional disease surveillance methods. A wastewater surveillance program should provide rapid turn around for sample measurements (ideally within 24 hours), but storage of samples is necessary for a variety of reasons including biobanking. Here we investigate how storage of wastewater solids at 4 °C, −20 °C, and −80 °C affects measured concentrations of SARS-CoV-2 RNA. We find that short term (7 or 8 d) storage of raw solids at 4 °C has little effect on measured concentrations of SARS-CoV-2 RNA, whereas longer term storage at 4 °C (35–122 d) or freezing reduces measurements by 60%, on average. We show that normalizing SARS-CoV-2 RNA concentrations by concentrations of pepper mild mottle virus (PMMoV) RNA, an endogenous wastewater virus, can correct for changes during storage as storage can have a similar effect on PMMoV RNA as on SARS-CoV-2 RNA. The reductions in SARS-CoV-2 RNA in solids during freeze thaws is less than those reported for the same target in liquid influent by several authors.

# INTRODUCTION

SARS-CoV-2 RNA in settled solids from wastewater treatment plants correlates to COVID-19 incidence in the sewershed population (*Wolfe et al., 2021*; *Graham et al., 2021*; *Peccia et al., 2020*; *D'Aoust et al., 2021*; *Kitamura et al., 2021*). As a result, local and federal governmental agencies are establishing wastewater-based epidemiology methods to help inform pandemic response (*United States Center for Disease Control, 2020*). SARS-CoV-2 RNA concentrations are measured in the wastewater and information about the incidence of COVID-19 can be inferred from the measurement concentrations (*Wolfe et al., 2021*; *Graham et al., 2021*; *Peccia et al., 2020*; *D'Aoust et al., 2021*; *Kitamura et al., 2021*).

Wastewater consists of liquid and solid components. While many wastewater surveillance efforts have focused on measuring SARS-CoV-2 RNA in the liquid component of wastewater (*Medema et al., 2020*; *Weidhaas et al., 2021*), the solids have $10^3$ to $10^4$ higher

Corresponding author
Alexandria B. Boehm,
aboehm@stanford.edu

concentrations of SARS-CoV-2 RNA on a per mass basis (*Graham et al., 2021*; *Li et al., 2021*). Settled solids are readily collected from the primary clarifier where they settle as part of the wastewater treatment process (often referred to as sludge), or they can be settled from wastewater influent using standard method SM2540 F (*AWWA, 2005*) if a wastewater treatment plant does not have a primary clarifier unit process.

In order for wastewater data on SARS-CoV-2 RNA concentrations to be useful for wastewater based epidemiology, real time disease response, samples should be analyzed quickly and results reported as soon as possible to public health officials. In such a scenario, samples are processed within hours of collection. However, sample storage remains essential. Samples may need to be stored for extended periods of time during transport or shipment from wastewater treatment plants to laboratories if the distance between them is far. In cases where a laboratory instrument malfunctions or results do not pass quality control metrics, samples might need to be rerun. Samples therefore need to be stored for at least as long as it takes to obtain laboratory results. Additionally, labs may want to create a biobank of samples; these samples can be used in the future to probe the presence of variants of concerns or other pathogens as needed. However, little is known about how sample storage affects the quantification of SARS-CoV-2 RNA in wastewater. If sample storage significantly changes the concentration of SARS-CoV-2 RNA relative to fresh, unstored samples, then measurements from stored samples may provide incorrect information on the incidence of COVID-19 in the sewersheds. The goal of this study is to provide data to help fill this knowledge gap.

Several studies have investigated how storage conditions affect quantification of SARS-CoV-2 RNA in wastewater to be used for wastewater based epidemiology, but most have focused on liquid influent (*Weidhaas et al., 2021*; *Whitney et al., 2021*; *Hokajärvi et al., 2021*; *Bivins et al., 2020*; *Ahmed et al., 2020*) and determined that storage and freeze thaws of the liquid influent can reduce measured concentrations of the viral RNA an order of magnitude or more. Only one study has examined persistence in wastewater solids: *Hokajärvi et al. (2021)* examined degradation of SARS-CoV-2 RNA in solids pelleted from raw influent *via* centrifugation. The goal of this study is to assess the impact of different realistic storage conditions on the quantification of SARS-CoV-2 RNA and an endogenous viral control (pepper mild mottle virus, PMMoV) in wastewater settled solids (also referred to as primary settled solids or sludge). PMMoV is highly abundant in human stool and domestic wastewater globally (*Colson et al., 2010*; *Rosario et al., 2009*). The results of this study will inform optimal storage conditions for settled solids for use for wastewater-based epidemiology.

## MATERIALS & METHODS

### Schematic overview of the experimental approach

We collected settled solids from wastewater treatment plants and analyzed them immediately within 6 hours to measure SARS-CoV-2 RNA concentrations and PMMoV RNA concentrations. The samples were then stored under different conditions (4 °C, −20 °C, or −80 °C) for different amounts of time and the SARS-CoV-2 RNA was
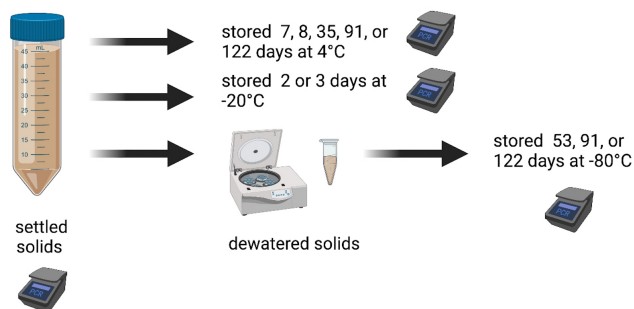

**Figure 1 Schematic of experiment.** We collected settled solids from wastewater treatment plants and analyzed them immediately within 6 h to measure SARS-CoV-2 RNA concentrations and PMMoV RNA concentrations. The samples were then stored under different conditions (4 °C, −20 °C, or −80 °C) for different amounts of time and the SARS-CoV-2 RNA was measured again using the same analytical methods. The samples stored at −80 °C were first dewatered prior to storage. Created with BioRender.com.

**Table 1 Samples used in this study.** Samples used in this study as well as the wastewater plant they were collected from (all located in California) and the date of sample collection, and the storage temperature and duration for the treatment applied to it. In order to protect the privacy of the plants and the inhabitants of their sewersheds, the precise locations are not provided herein.

| Sample ID | Plant | Date | Storage condition(s) | Duration(s) of storage |
|---|---|---|---|---|
| SJ200080 | SJ | 2/28/21 | 4 °C | 8 d |
| D300230 | D | 3/1/21 | 4 °C | 7 d |
| OSP700097 | Ocean | 3/1/21 | 4 °C | 7 d |
| SJ200082 | SJ | 2/23/21 | −20 °C | 3 d |
| D300234 | D | 2/24/21 | −20 °C | 2 d |
| OSP700126 | Ocean | 2/24/21 | −20 °C | 2 d |
| SAC900313 | SAC | 3/1/21 | 4 °C/−20 °C | 7 d/2 d[a] |
| SJ200128 | SJ | 2/25/21 | 4 °C/−80 °C | 35 d |
| SJ20023 | SJ | 11/30/20 | 4 °C/−80 °C | 122 d |
| D300243 | D | 2/25/21 | 4 °C/−80 °C | 35 d |
| D300066 | D | 12/31/20 | 4 °C/−80 °C | 91 d |

**Notes.**
[a] The sample stored at −20 °C was stored for 2 d and the sample stored at 4 °C was stored for 7 d.

measured again using the same analytical methods. The samples stored at −80 °C were first dewatered prior to storage. The measurements obtained immediately from the "fresh" sample (the "control") and measurements obtained from the same sample after storage (called "treatment") were then compared to assess how storage treatments affect the concentration of SARS-CoV-2 and PMMoV RNA. A schematic of the experimental approach is shown in Fig. 1, and outlined in Table 1. Full details are provided below.

## Sample collection

Eleven (11) 50-ml samples of settled solids were collected from the primary clarifiers at four unique wastewater treatment plants (Table 1) using sterile technique and clean

containers. Samples were immediately stored on ice and transported to the lab where they were processed within 6 hours of sample pick up from the plants with high throughput methods (*Topol et al., 2021a*; *Topol et al., 2021b*; *Topol et al., 2021c*). These un-stored samples are referred to as "controls". Thereafter, the samples were subjected to different storage treatments in the laboratory (Table 1). Raw samples were either stored at 4 °C for 7, 8, 35, 91 or 122 days, or -20 °C for 2 or 3 d. Four samples were stored as dewatered solids (described below) at −80 °C for 35, 91 or 122 days. The lengths and temperatures of storage were chosen to represent the range of conditions under which we may need to store samples for our own wastewater surveillance project. After storage, refrigerated samples were immediately processed, and frozen samples were removed from freezers and defrosted at 4 °C for 24 h and then processed according to Topol et al. (*Topol et al., 2021a*; *Topol et al., 2021b*; *Topol et al., 2021c*). Stored samples are referred to as "treatments". In total, there were 16 samples subjected to a storage treatment.

## Sample preparation

The solids were dewatered by centrifugation at 24,000 × g for 30 minutes at 4 °C and aspirating and discarding the supernatant. A 0.5–1 g aliquot of the dewatered solids was dried at 110 °C for 19–24 hrs to determine its dry weight. Bovine coronavirus (BCoV) was used as a positive recovery control. Each day, attenuated bovine coronavirus vaccine (PBS Animal Health, Calf-Guard Cattle Vaccine) was spiked into DNA/RNA shield solution (Zymo Research, Irvine, CA, Cat # R1100) at a concentration of 1.5 μL /mL. Dewatered solids were resuspended in the BCoV-spiked DNA/RNA shield to a concentration of 75 mg/mL. This concentration of solids was chosen as previous work titrated solutions with varying concentrations of solids to identify a concentration at which inhibition of the SARS-CoV-2 assays was minimized. Five to 10 5/32″ Stainless Steel Grinding Balls (OPS Diagnostics, Lebanon, NJ, GBSS 156-5000-01) were added to each sample which was subsequently homogenized by shaking with a Geno/Grinder 2010 (Spex SamplePrep, Metuchen, NJ). Samples were then briefly centrifuged to remove air bubbles introduced during the homogenization process, and then vortexed to re-mix the sample.

## RNA extraction

RNA was extracted from 10 replicate aliquots per sample. For each replicate, RNA was extracted from 300 μl of homogenized sample using the Chemagic™ Viral DNA/RNA 300 Kit H96 (Perkin Elmer, Akron, OH, Cat # CMG-1033 S T) for the Chemagic 360 (Perkin Elmer, Akron, OH) followed by PCR Inhibitor Removal with the Zymo OneStep-96 PCR Inhibitor Removal Kit (Cat # D6030). Extraction negative controls (water) and extraction positive controls (500 copies of SARS-CoV-2 genomic RNA (ATCC® VR-1986D™)) in the BCoV-spiked DNA/RNA shield solution described above) were extracted using the same protocol as the homogenized samples.

## Digital PCR

RNA extracts were used as template in digital droplet RT-PCR assays for SARS-CoV-2 N, S, and ORF1a RNA gene targets in a triplex assay, and BCoV and PMMoV in a duplex assay (see Table 2 for primer and probe sequences, purchased from Integrated DNA
**Table 2  The molecular targets used in this study as well as the primer and probe sequences.** The N, S, and ORF1a genes are located within SARS-CoV-2 genome. The BCoV target is for bovine coronavirus, a process control spiked into the sample during processing. PMMoV is for the internal endogenous control which is naturally present in high concentrations in the samples. Additional details of these assays can be found in *Huisman et al. (2021)*.

| Target | Primer/Probe | Sequence |
|---|---|---|
| N Gene | Forward | CATTACGTTTGGTGGACCCT |
|  | Reverse | CCTTGCCATGTTGAGTGAGA |
|  | Probe | CGCGATCAAAACAACGTCGG (5′FAM/ZEN/3′IBFQ) |
| S Gene | Forward | CAGACTAATTCTCCTCGGCG |
|  | Reverse | TGCACCAAGTGACATAGTGT |
|  | Probe | AGCTAGTCAATCCATCATTGCCT (5′HEX/ZEN/3′IBFQ) |
| ORF1a | Forward | CAGAACTGGAACCACCTTGT |
|  | Reverse | TACAGTTGAATTGGCAGGCA |
|  | Probe | TGCCACAGTACGTCTACAAGC (5′ FAM or HEX/ZEN/3′ IBFQ) |
| BCoV | Forward | CTGGAAGTTGGTGGAGTT |
|  | Reverse | ATTATCGGCCTAACATACATC |
|  | Probe | CCTTCATATCTATACACATCAAGTTGTT (5′ FAM/ZEN/3′ IBFQ) |
| PMMoV | Forward | GAGTGGTTTGACCTTAACGTTTGA |
|  | Reverse | TTGTCGGTTGCAATGCAAGT |
|  | Probe | CCTACCGAAGCAAATG (5′ HEX/ZEN/3′ IBFQ) |

Technologies). Undiluted extract was used for the SARS-CoV-2 assay template and a 1:100 dilution of the extract was used for the BCoV / PMMoV assay template. Digital RT-PCR was performed on 20 μl samples from a 22 μl reaction volume, prepared using 5.5 μl template, mixed with 5.5 μl of One-Step RT-ddPCR Advanced Kit for Probes (Bio-Rad, Hercules, CA Cat # 1863021), 2.2 μl Reverse Transcriptase, 1.1 μl DTT and primers and probes at a final concentration of 900nM and 250nM respectively. Droplets were generated using the AutoDG Automated Droplet Generator (Bio-Rad). PCR was performed using Mastercycler Pro with the following protocol: reverse transcription at 50 °C for 60 minutes, enzyme activation at 95 °C for 5 minutes, 40 cycles of denaturation at 95 °C for 30 seconds and annealing and extension at either 59 °C (for SARS-CoV-2 assay) or 56 °C (for PMMoV/BCoV duplex assay) for 30 seconds, enzyme deactivation at 98 °C for 10 minutes then an indefinite hold at 4 °C. The ramp rate for temperature changes were set to 2 °C/second and the final hold at 4 °C was performed for a minimum of 30 minutes to allow the droplets to stabilize. Droplets were analyzed using the QX200 Droplet Reader (Bio-Rad). All liquid transfers were performed using the Agilent Bravo (Agilent Technologies, Palo Alto, CA).

Each sample was run in 10 replicate wells, extraction negative controls were run in 7 wells, and extraction positive controls in 1 well. In addition, PCR positive controls for SARS-CoV-2 RNA, BCoV, and PMMoV were run in 1 well, and NTC were run in 7

wells. Positive controls consisted of BCoV and PMMoV gene block controls (purchased from Integrated DNA Technologies) and gRNA of SARS-CoV-2 (ATCC® VR-1986D™). Results from replicate wells were merged for analysis. Thresholding was done using QuantaSoft™ Analysis Pro Software (Bio-Rad, version 1.0.596). Additional details are provided in supporting material per the dMIQE guidelines (*The dMIQE Group & Huggett, 2020*).

## Data analysis

Concentrations of RNA targets were converted to concentrations per dry weight of solids in units of copies/g dry weight. The total error is reported as 68% confidence intervals and includes the errors associated with the Poisson distribution and the variability among the 10 replicates. The recovery of BCoV was determined by normalizing the concentration of BCoV by the expected concentration given the value measured in the spiked DNA/RNA shield. If the BCoV recovery was less than 10%, then the sample was rerun.

PMMoV, N, S, ORF1a, as well as N/PMMoV, S/PMMoV, and ORF1a/PMMoV were compared for each sample control and treatment, 1 by 1, by examining the measurement and the 68% error associated with the measurement. The error associated with the quotients was estimated by propagating errors on the numerator and denominator, as described by *Graham et al. (2021)*. In brief, the relative error of the quotient is approximated by the square root of the sum of the relative errors of the numerator and denominator squared. If the measurement of the treatment condition fell within the error range of the control condition, then the measurement was deemed "not different". This approach is equivalent to a $t$-test where the null hypothesis (Ho) is the value of the treatment is the same as the control and the alternate hypothesis (Ha) is that the values are different. In this study, we are particularly concerned about type 2 errors (failing to reject Ho when it is false) as we are concerned with whether storage renders different measurements. As such, in order to increase the power of the analysis, we chose to make comparisons using the 68% confidence intervals. With the 10 replicates, this gives ~90% power of avoiding a type 2 error assuming an effect size equal to the standard deviation. Note that if we had used 95% confidence intervals in our analysis rather than 68% confidence intervals, the likelihood of not detecting a difference in the treatments and controls would be higher as 95% confidence intervals are approximately two times larger than 68% confidence intervals.

For measurements deemed "different", the percent difference (% diff) was calculated as % diff $= 100 \times$ (control-treatment)/control where control and treatment are the associated measurements. A positive % diff indicates that the treatment result is smaller than the control, whereas a negative percent indicates the treatment had higher concentrations than the control. Errors for % diff were propagated from the measurements as standard deviations.

# RESULTS

## Quality control

All positive extraction and PCR controls were positive, and all negative extraction controls and negative PCR controls were negative indicating no cross contamination

between samples. Recovery of spiked BCoV was above 10% for all samples so no samples were rerun owing to unacceptable recovery. SARS-CoV-2 and PMMoV targets were detected in all samples. Data from the experiments is available through Stanford Digital Repository (*Simpson & Boehm, 2021*).

### Short-term storage at 4 °C

Two experiments were carried out to investigate the effects of storage on 8 samples of raw settled solids samples at 4 °C (Table 1). First, all samples were processed within 6 hours of collection and the resultant measurements were treated as those of the control condition. Then, in one experiment, 4 samples were stored for 7–8 days at 4 °C prior to being processed a second time; and in the other, 4 samples were stored between 35 days and 122 days prior to being processed a second time. The stored conditions are referred to as treatments.

SARS-CoV-2 RNA measurements made after 7–8 d of storage at 4 °C were not different from the control condition for 3 of the 4 samples; PMMoV was not different between treatment and control for all 4 of the samples. When SARS-CoV-2 RNA gene concentrations were normalized by PMMoV gene concentrations, the ratios were not different between treatment and control in any of the 4 samples (Fig. 2). For the sample that had lower SARS-CoV-2 RNA gene concentrations in the treatment compared to the control, the concentrations of the N, S, and ORF1 genes differed by 60%, 80%, and 73% respectively (Table 3).

### Long-term storage at 4 °C

Longer storage at 4 °C (between 35 and 122 days) resulted in significantly lower measurements of SARS-CoV-2 RNA in 3 of the 4 tested samples, and lower PMMoV RNA concentrations in 2 of 4 samples. Normalizing SARS-CoV-2 RNA concentrations by PMMoV concentrations "corrected" the differences observed in 1 sample, but not the other 2. For those 2 samples, ratios were lower and different in the treatments compared to the controls (Fig. 2). The difference between the treatment and controls are shown in Table 3. The average difference in the measurements, when they were different, was 44%.

### Storage at −20 °C

Four samples were processed within 6 h of collection to obtain control measurements. The same samples were also frozen at −20 °C for 2 or 3 days, and then defrosted and processed to obtain treatment measurement (Table 1).

SARS-CoV-2 RNA concentrations in treatments were different in 3 of the 4 samples compared to controls (lower in treatment *versus* control in 1 and higher in 2, average difference 89%). PMMoV concentrations were not different between treatments and controls. Normalizing SARS-CoV-2 RNA concentrations by PMMoV RNA concentrations, "corrected" the differences in SARS-CoV-2 RNA observed in 2 of the 3 samples (Fig. 3); for the single sample where the ratio of SARS-CoV-2 RNA to PMMoV RNA concentrations was different, the treatment had 61% (average) lower ratio than the control . Differences between treatments and controls are summarized in Table 3.

Simpson et al. (2021), *PeerJ*, DOI 10.7717/peerj.11933

**Table 3  Results for comparisons between experimental treatments and controls.** The percent difference (% diff) in SARS-CoV-2 RNA measured in treatments versus their control is shown then the difference was significantly different. "N" indicates that measurements or ratios were not different. The value after the ± is the standard deviation propagated from the measurements used to make the calculation. A positive percent difference indicates the treatment was lower than the control, a negative percent difference indicates the treatment was higher in the control. See methods for more details on the calculations.

| Sample | Treatment | % diff N | % diff S | % diff ORF1a | % diff PMMoV | % diff N/PMMoV | % diff S/PMMoV | % diff ORF1a/PMMoV |
|---|---|---|---|---|---|---|---|---|
| SJ200080 | 4 °C/8 d | N | N | N | N | N | N | N |
| D300230 | 4 °C/7 d | N | N | N | N | N | N | N |
| SAC900313 | 4 °C/7 d | N | N | N | N | N | N | N |
| OSP700097 | 4 °C/7 d | 60 ± 48 | 80 ± 49 | 73 ± 67 | N | N | N | N |
| SJ200128 | 4 °C/35 d | 49 ± 22 | 49 ± 19 | 59 ± 19 | 45 ± 15 | N | N | N |
| D300243 | 4 °C/35 d | 42 ± 23 | 52 ± 31 | 55 ± 22 | N | 40 ± 41 | 50 ± 48 | 53 ± 49 |
| SJ200023 | 4 °C/122 d | 55 ± 16 | 58 ± 12 | 63 ± 14 | 29 ± 10 | 36 ± 27 | 41 ± 23 | 48 ± 26 |
| D300066 | 4 °C/91 d | N | N | N | N | N | N | N |
| SJ200082 | −20 °C/3 d | 58 ± 21 | 57 ± 18 | 46 ± 28 | N | 65 ± 32 | 64 ± 29 | 54 ± 38 |
| D300234 | −20 °C/2 d | N | −100 ± 56 | −170 ± 35 | N | N | N | N |
| SAC900313 | −20 °C/2 d | −86 ± 28 | −71 ± 38 | N | N | N | N | N |
| OSP700126 | −20 °C/2 d | N | N | N | N | N | N | N |
| SJ200128 | −80 °C/35 d | 76 ± 21 | 62 ± 17 | 79 ± 18 | 79 ± 18 | N | N | N |
| D300243 | −80 °C/35 d | 40 ± 21 | 47 ± 27 | 59 ± 19 | 21 ± 13 | N | 33 ± 40 | 48 ± 33 |
| SJ200023 | −80 °C/122 d | 89 ± 13 | 86 ± 9 | 89 ± 10 | 74 ± 13 | 57 ± 32 | 48 ± 31 | 59 ± 29 |
| D300066 | −80 °C/91 d | 75 ± 38 | 70 ± 42 | 78 ± 51 | 49 ± 34 | N | N | N |

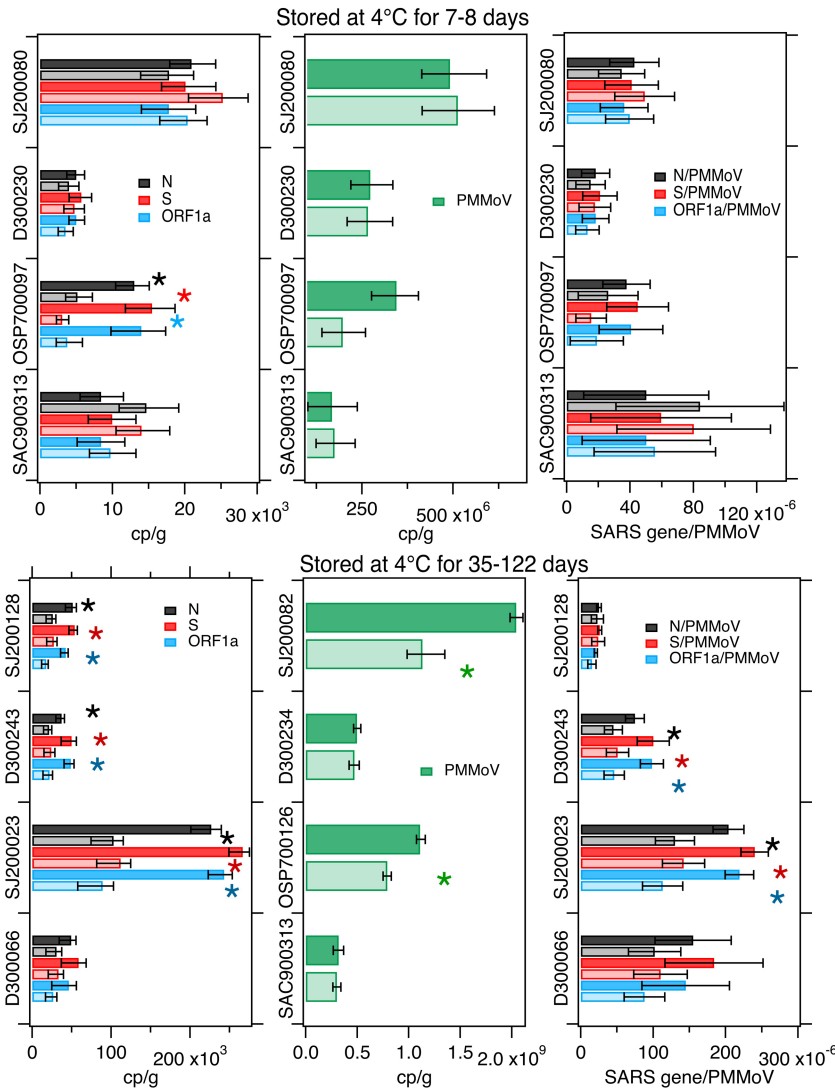

**Figure 2 Measurements in 4 °C treatments and controls.** Concentrations of SARS-CoV-2 RNA targets (N, S, ORF1a, left column) and PMMoV RNA (middle column), as well as their ratios (right column) as measured in settled solids in controls (darker bars) and treatments (lighter bars) and total errors as reported by the digital PCR instrument, or in the case of the ratios, the errors were propagated. Units are copies per gram dry weight (cp/g) for SARS-CoV-2 and PMMoV RNA targets, and there are no units for the ratio. Asterisks denote measurements or calculated ratios for which the errors on the treatments and controls do not overlap and indicate the measurements were different. Top row shows experiments where solids were stored at 4 °C for 7–8 days while the bottom row shows experiments where solids were stored at 4 °C for 35-122 days (see Table 1). The sample names on the y-axes consist of a letter indicating the plant location and a random number. The scales are different between plots to properly allow the plotting of the values which vary between samples.

## Storage at −80 °C

Four samples were processed within 6 hours of collection to obtain control measurements and also dewatered, frozen at −80 °C for between 35 and 122 days, and then defrosted and processed to obtain treatment measurements (Table 1).

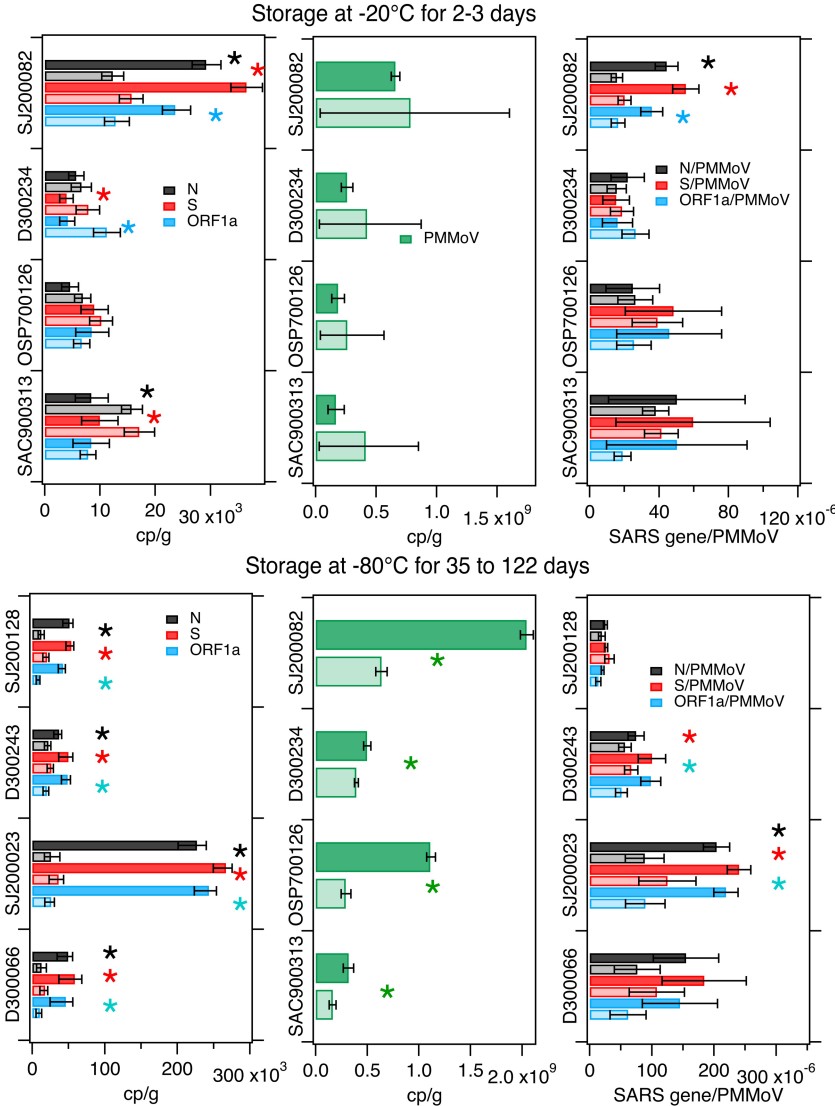

**Figure 3  Comparison on measurements in freeze/thaw treatments and controls.** Concentrations of SARS-CoV-2 RNA targets (N, S, ORF1a, left column) and PMMoV RNA (middle column), as well as their ratios (right column) as measured in settled solids in controls (darker bars) and treatments (lighter bars) and total errors as reported by the digital PCR instrument, or in the case of the ratios, the errors were propagated. Units are copies per gram dry weight (cp/g) for SARS-CoV-2 and PMMoV RNA targets, and there are no units for the ratio. Asterisks denote measurements or calculated ratios for which the errors on the treatments and controls do not overlap and indicate the measurements were different. Top row shows experiments where solids were stored at −20 °C for 2–3 days while the bottom row shows experiments where dewatered solids were stored at −80 °C for 35–122 days (see Table 1).

SARS-CoV-2 and PMMoV RNA concentrations were statistically lower in the treatments compared to the controls by 40% to 90% (Fig. 3 and Table 3). Normalizing SARS-CoV-2 RNA concentrations by PMMoV concentrations "corrected" the differences for 2 of the 4 samples. For the other 2, the ratio of SARS-CoV-2 RNA to PMMoV RNA concentrations was 49% (on average) lower in the treatment than the control.

### Overview of combined results

Across the treatment/samples where we observed a change in SARS-CoV-2 RNA concentrations relative to the control ($n = 11$ of 16), there was a reduction in measured SARS-CoV-2 RNA concentration in all but 2 sample treatments (mean percent reduction of 63% $\pm14\%$ standard deviation, $n = 27$: 9 treatment/samples $\times$ 3 SARS-CoV-2 gene measurements different per treatment/sample). In the 2 cases where we observed an increase in SARS-CoV-2 concentrations in the treatment compared to the control, that increase was 107% $\pm$ 44% on average ($n = 4$, 2 sample /treatments $\times$ 2 SARS-CoV-2 genes were different per sample/treatment). All measurements made in samples receiving a treatment were within an order of magnitude of the measurement in the control.

## DISCUSSION

SARS-CoV-2 RNA concentrations in wastewater settled solids correlate to COVID-19 incidence in surrounding sewersheds (*Wolfe et al., 2021*; *Graham et al., 2021*; *Peccia et al., 2020*; *D'Aoust et al., 2021*; *Kitamura et al., 2021*). As a result, SARS-CoV-2 RNA concentrations in wastewater settled solids are being used to inform the response to the COVID-19 pandemic. While immediate processing of samples after they are collected is ideal, it is necessary to store and archive samples in case samples need to be rerun due to failed quality control metrics, or if additional analyses are needed to investigate the presence of other viruses or viral variants, for example. There is limited data on how storage of wastewater samples affects the concentration of SARS-CoV-2 RNA in the samples, so we carried out experiments to help fill this knowledge gap. Our experiments suggest that storage of wastewater settled solids samples, either refrigerated or frozen, may change the concentrations of SARS-CoV-2 RNA measured in the samples, but changes are less than an order of magnitude even for samples stored over 100 days.

Several studies have examined the effect of sample storage on SARS-CoV-2 RNA quantification in wastewater influent which represents a different matrix than that examined herein. Influent consists primarily of liquid wastewater rather than solids. *Hokajärvi et al. (2021)* examined the effect of storage on detection of SARS-CoV-2 RNA in liquid influent and found small differences resulting from storage at freezing temperatures and first order decay of the RNA in influent stored at 4 °C with a $T_{90}$ (time until 1 $\log_{10}$ reduction of concentration) of 36 to 52 days depending on the genomic target. *Ahmed et al. (2020)* report similar results as Hokajärvi et al. for decay of the SARS-CoV-2 targets in influent during storage at 4 °C. *Markt et al. (2021)* found minimal differences in SARS-CoV-2 RNA concentrations measured in influent stored for up to 7 days at 4 °C compared to concentrations measured with no storage, but found more than an order of magnitude decrease in SARS-CoV-2 RNA concentrations in samples that were stored frozen and subject to a freeze thaw. *Fernandez-Cassi et al. (2021)* report extensive reduction in SARS-CoV-2 RNA concentrations measured in liquid wastewater stored at 4 °C and $-20$ °C (1–2 orders of magnitude). Our results regarding wastewater solids stored at 4 °C are similar to those presented in these influent studies, except for *Fernandez-Cassi et al. (2021)*; overall we saw minimal reduction (less than an order of magnitude) even for

samples stored over 100 days. However, the effect of freeze thaw on our measurements with solids is small compared to those reported by *Markt et al. (2021)* and *Fernandez-Cassi et al. (2021)* for infuent. We could identify only one published study on SARS-CoV-2 RNA decay in solids: *Hokajärvi et al. (2021)* report minimal decay (between 0% and 20%) of SARS-CoV-2 RNA solids pelleted rom raw influent by centrifugation during storage at 4 °C, −20 °C, and −75 °C for 84 days, within the range of results reported herein.

Researchers have used PMMoV RNA as an internal process control in their efforts to monitor SARS-CoV-2 RNA in wastewater (*Wolfe et al., 2021*; *Wu et al., 2020*; *Feng et al., 2021*). Assuming endogenous PMMoV RNA is recovered in the sample processing and RNA extraction and purification process in the same manner as SARS-CoV-2 RNA, then normalizing SARS-CoV-2 RNA by PMMoV RNA provides a ratio that does not depend on recovery. *Wolfe et al. (2021)* showed the ratio of SARS-CoV-2 RNA/PMMoV RNA in settled solids is associated with COVID-19 incidence rates empirically, and the relationship between the ratio and COVID-19 incidence rates also falls from a mass balance model that relates wastewater solid concentrations to the number of people shedding SARS-CoV-2 RNA in their stool.

In this study, we found that normalizing SARS-CoV-2 RNA by PMMoV RNA concentrations corrected for changes in concentration that may result during storage. In 7 of the 11 sample/treatments that showed differences between SARS-CoV-2 RNA concentrations and the control, SARS-CoV-2/PMMoV was not different between treatment and control. When there were differences, they were less than an order of magnitude. PMMoV RNA concentration was often affected by storage in a similar manner as SARS-CoV-2 RNA concentrations, thus highlighting an additional benefit of using the internal control to interpret concentrations of the SARS-CoV-2 RNA targets in wastewater-based epidemiology applications. The ability to effectively correct for the impact of storage on samples suggests that the primary concerns for sample storage of wastewater solids are related to times when SARS-CoV-2 concentrations are nearing the limit of detection, and during periods of low incidence immediate sample processing should be a higher priority.

Additional research should examine a time course of decay of SARS-CoV-2 RNA for a single solids sample stored for various lengths of time, and also investigate the effects of multiple freeze thaws on target quantification. Work with additional viral targets may also be useful to provide guidance on storage for wastewater-based epidemiology applications beyond COVID-19 including using RNA from gastro-intestinal viruses like norovirus and rotavirus to infer incidence of diarrheal illnesses. Finally, our study was not powered to investigate whether storage affected viral quantification in wastewater solids from different wastewater treatment plants in different ways; we used samples from diverse plants to capture potential variations between properties of wastewater solids. Additional work should investigate if storage has differential effects on samples from different wastewater treatment plants. Importantly, studies on the effect of storage on quantification of wastewater based epidemiology targets in wastewater influent and solids should be carried out in the near term, before there is a pressing need to use the measurements for disease outbreak response.

## CONCLUSIONS

Wastewater based epidemiology is not a new field. It has previously been used to surveil populations for infectious diseases including polio (*Brouwer et al., 2018*), hepatitis (*McCall et al., 2020*), and salmonellosis (*Diemert & Yan, 2019*). Its wide-spread use globally during the COVID-19 pandemic to infer trends in COVID-19 incidence, however, is unprecedented. As such, new research is needed to continue to fill research gaps and develop models that link wastewater concentrations of pathogens to disease incidence rates.

One of those research gaps concerns storage of wastewater samples, and how storage affects measured concentrations of wastewater based epidemiology targets –in this case SARS-CoV-2 RNA. Storage of wastewater solids for use in wastewater-based epidemiology applications is essential. Here we examined how storage at 4 °C for short (7–8 d) and long durations (35–122 d), −20 °C for short (2–3 d), and −80 °C for long durations (35–122 d) affects SARS-CoV-2 RNA measurements in wastewater solids, and whether normalizing measurements by concentrations of an internal process control corrects for the effects of storage.

We found storage at 4 °C for short durations of 7–8 days had limited to no effect on measured concentrations, but other storage conditions and durations affected concentrations by reducing them by 61%, on average, and in one case increasing them by up to 170%. However, we found that the normalizing concentrations by the internal process control PMMoV helped to correct for the observed differences.

Degradation of wastewater based epidemiology targets during storage represents a challenge to application of the tool for public health responses to infectious disease outbreaks. Storage of samples is essential. Identifying matrices and storage conditions where there is limited degradation is essential. With this in mind, we recommend short duration storage at 4 °C, and normalizing concentrations of SARS-CoV-2 RNA by concentrations of PMMoV RNA in the sample for use in models that relate trends in wastewater to disease incidence. Even under the longer storage conditions including those that required a freeze/thaw, changes in concentrations observed with the solids were less than one order of magnitude and similar among samples subjected to the same treatment.

## ACKNOWLEDGEMENTS

We acknowledge the wastewater treatment plant staff for providing the samples. This study was performed on the ancestral and unceded lands of the Muwekma Ohlone people. We pay our respects to them and their Elders, past and present, and are grateful for the opportunity to live and work here.

### Funding

This work was funded by the CDC-Foundation. The funders had no role in study design, data collection and analysis, decision to publish, or preparation of the manuscript.

## Grant Disclosures

The following grant information was disclosed by the authors:
CDC-Foundation.

## Competing Interests

Adrian Simpson, Aaron Topol and Bradley White are employed by Verily Life Sciences.

## Author Contributions

- Adrian Simpson conceived and designed the experiments, performed the experiments, analyzed the data, prepared figures and/or tables, and approved the final draft.
- Aaron Topol conceived and designed the experiments, performed the experiments, authored or reviewed drafts of the paper, and approved the final draft.
- Bradley J. White, Marlene K. Wolfe and Krista R. Wigginton conceived and designed the experiments, authored or reviewed drafts of the paper, and approved the final draft.
- Alexandria B. Boehm conceived and designed the experiments, analyzed the data, prepared figures and/or tables, and approved the final draft.

## Data Availability

Simpson, Adrian and Boehm, Alexandria. (2021). Effect of storage on concentrations of SARS-CoV-2 RNA in settled solids of wastewater treatment plants. Stanford Digital Repository. Available at: https://purl.stanford.edu/yn042kx5009.

## Supplemental Information

Supplemental information for this article can be found online at http://dx.doi.org/10.7717/peerj.11933#supplemental-information.

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
