# Peer review of "Effect of storage conditions on SARS-CoV-2 RNA quantification in wastewater solids"

_PeerJ, doi:10.7717/peerj.11933_

## Round 0.1 · original submission · Major Revisions

Authors, please kindly attend to comments raised by reviewers. You can see there is a wide range of comments (from major revisions, to accept), which is the joy of peer review. Please, provide details where possible. Look forward to your revised manuscript. Thank you.

Reviewer 1 ·

Basic reporting

The manuscript is concise and well written. There are a few typos. Regarding the background/context I have two comments:

L56 More nuance is warranted here. A few of these studies used both the liquid and solids fraction of the influent together while another in Helsinki separated the two and then spiked. This is mentioned in the discussion but not here in the introduction. It would be useful to better describe the sample types used in previous studies.

L59 Per my reading, the study from Helsinki considered signal degradation in the solids fraction (separation by centrifugation) of influent samples.

Experimental design

This is a well designed and executed study and represents excellent work by the authors. The inclusion of MIQE checklist is also appreciated. One comment:

L137 The method of propagating error should be detailed further or incorporated by reference (maybe similar to Graham et al.?)

Validity of the findings

This is perhaps where I only have a real criticism/contribution to suggest:

L142 The justification for using a 68% confidence interval is compelling. But why not perform the analysis with both 68% and 95% intervals? Would all the significant differences remain? The authors should consider performing that analysis at the 95% confidence level and including those results in the SI. If the results are consistent, this would greatly strengthen the inferences. If they are not, then the authors can use their logic to justify their conclusions based on the type II error.

A few other comments pertinent to the discussion:

L228 Hokajärvi also examined solids (Table 2) that were pelleted via centrifugation and then spiked. It is worth discussing the current results relevant to their observations.

L243 This statement contradicts L59 unless the authors are making a distinction between solids separated by centrifugation and “settled solids”.

L244 Would the results of the current study be similar to those in Helsinki if 95% confidence interval was used instead of 68%?

Reviewer 2 ·

Basic reporting

The paper is well written using clear professional English. A sufficient knowledge gap has been established. The article structure is good. Figures are of publication quality.

Experimental design

The experimental design is good and contains references.

Validity of the findings

No comments.

Additional comments

No Comments.

Reviewer 3 ·

Basic reporting

The language of the article is good. A few instances of things not being too clear or incorrect are pointed out below in specific comments. Some figures have misaligned graphs within them (see specific comment). Results section must be significantly reworked.

Here are some specific comments:

Lines 29-30: Is PMMoV a wastewater virus? Plant virus quasi-ubiquitous in sanitary sewage?

Line 36: Two of the 3 citations are by authors however several groups have outlined this, it may be good to cita a few more pieces of litterature: Kitamura (2021), Kocamemi (2020), D'Aoust (2021), Chik (2021). Balboa also published their preprint after you submitted (2021) it would be nice to acknowledge them during the review.

Line 48-49: "In such a scenario, samples are not stored, but are processed as soon as they are collected. Even if this is done as recommended, sample storage is essential." I think this paragraph should be reworked a bit. I can see what the authors are trying to do and justify why storage is important, but the result reads a bit clumsy. You could write that storage is important because of transport time, laboratories for analysis may not be located in the direct vicinity of the sampling sites (and so must be shipped), and bio-banks and reruns are sometimes necessary.

Results:

Lines 163-169, this sounds like methods... should this be in the methods to explain the experimental procedure? It seems like now this is a breakdown of the methods, but it's not results. This section should be moved to methods, and it's hard to textually understand what was done. Presenting the breakdown of when samples are testing in a table format would immensely help.

Line 185: ~50% An actual value would be very helpful here.

Figure 1: Would it be possible to provide more textually descriptive names for the samples. The sample names as they are are obtuse. Graphs are misaligned with eachother (particularly bottom 3), units for the bars are a bit bizzare and are not the same while the targets are the same, hence do not really provide easy comparison. The graphs should have the same X axises.

The whole section needs to be tidied up and tightened. Approximations of changes in concentrations is not really acceptable if the methods and numbers are to be quantitative. Needs rework. The information is valuable and the graphs (aside from the axises) are doing a decent job at dissimenating the information, however the textual description of the results section need to be brought up to a higher standard seen at other places in the manuscript. The language is too vague.

Discussion:

Lines 217-224: this is what should be in the results. This type of language, the n's, etc. The discussion section should describe of the implications of the observed results however, not give more results.

Experimental design

The materials and methods section was reasonably well done. The experiment was well planned, however the planning part of it is not well communicated to the reader. Furthermore, some inconsistencies from section to section exist making it appear as though multiple people worked on the sections but some formatting and writing style/choice differences are very apparent. The section must be reworked it make it flow better as currently from section to section its inconsistent.

Here are some specific comments:

Line 68: Settled solids from the primary clarifiers - are we talking of primary sludge? Settled solids may not be the optimal term for wastewater people.

General comment for methods - please provide information such as manufacturer and part numbers for equipment, supplies, reagents, etc. It's there in the digital PCR section of the methods but not other sections. It's making it evident different sections were written by different people, or that these details were not standardized throughout the document.

Line 71: Without going in great detail, adding one sentence briefly describing the method would help a bit for the readability of the methods. I feel stating high-throughput methods alone may not be descriptive enough. Ex; what makes the method high throughput?

Line 73: From the text alone the methods imply that we would have several points in between the ranges of days (7-122), however when the reader gets to later sections it is shown that the samples were tested at 7 AND 122, with nothing in between. Having a "-" with a range can therefore be a bit misleading and actual experimental conditions should be stated, i.e. "stored at 4°C for 7, and 122 days). Furthermore, what was the rationale for the storage times selected?

Line 97: extra space before 500 copies and (.

Line 103: The information about being purchased from IDT could in my opinion be moved to Table 3. In addition, although it's pretty obvious for those of us in North America for international readers it may not be immediately apparent what/who IDT is. I think the company name (Integrated DNA Technologies) should be somewhere in the manuscript, either the first time you use the IDT acronym or as a footer for Table 3 explaining that probes and primers were purchased from IDT.

Lines 101-117: Whole section has different font for °C than the rest of the document. Please fix. Alt+248 (numpad) will do the correct degree symbol.

Some of the things that should have been in the material and methods section were found in the results section. This has been highlighted above in 1.

Discussion was significantly better written than the results section. Well done for that section!

Validity of the findings

Findings are probably valid but unfortunately with the results written the way they are currently it's not immediately possible for the reader to come to the same conclusions as the authors as the results are written as approximations (ie.: the
205 ratio of SARS-CoV-2 RNA to PMMoV RNA concentrations was ~50% lower in the treatment than the control. /// Differences between the experiments are shown in Table 3;
differences were between ~60% and 100%.). The section needs to be tightened significantly.

Additional comments

The manuscript was informative but the materials and methods and results sections need significant work. Some information that should be in M&M is in the results section. The results are given as vague approximations only.

The experimental protocol and outline of when samples were tested should be in a table format or in a format that's more easy for the reader to comprehend.

The discussion portion of the manuscript was significantly better quality than the materials and methods and the results. The manuscript is definitely worthy of publication as the information will be useful to the scientific community at large (and particularly those of us engaging in WBE) however those two sections must be reworked to bring them up to an acceptable standard.

Good job, and looking forward to seeing the final product published.

---

## Round 0.2 · Minor Revisions

Reviewers have commented on your work, and have recommended acceptance. One reviewer observed few other amendments, kindly consider them. In addition, the editor encourages authors to consider:

a) Introduction is not sufficient. Please, provide details on the import of storage conditions on wastewater treatment, why are they relevant to the context of this work? It appears the justification case is not strong, provide more evidence and a clearer gap which is to be filled by this study. What makes RNA quantification the approach of focus for this study, please make a stronger case for why it is being studied.

b) Materials and methods, given the nature of this work, please start this section with a subsection captioned 'Schematic overview of the experimental program', comprising 3-4 sentences and supported by a flow diagram. The idea here is to provide a snapshot of the experimental approach, how samples were collected, up to the point of analysis. Kindly make sure it is contextualised to the specific objective of this work. The purpose is NGR o guide readers, and draw them into appreciating and understanding this work better

Results and Discussion is ok. Authors are encouraged to strengthen the discussion, with more synthesised relevant literature.

Conclusions, it would be great if authors are able to brainstorm on (potential) challenges encountered. Important also is to strengthen the direction for future studies? Which areas are very useful for future research to undertake?

The editor will be looking forward to details on all the points raised here. Authors are encouraged to provide sufficient details. The work will be further improved with these.

Thank you very much

Reviewer 1 ·

Basic reporting

No comments. The manuscript is well written and ready for acceptance.

Experimental design

No further comments. I recommend for acceptance.

Validity of the findings

No further comments. I recommend for acceptance.

Additional comments

No further comments. I recommend for acceptance. Congratulations to the authors on well done study.

Reviewer 3 ·

Basic reporting

Changes made as requested. Very satisfied. Strong manuscript!

Experimental design

Changes done as required. Thank you!

Validity of the findings

Manuscript reads better. Thank you!

Additional comments

Great work, thank you for considering recommendations and applying them as required!

Few minor things to fix and you are good to go!

Line 240 of tracked changes doc: 107%, spacing between 107% and ±... to remain consistent with above.

Table 1, 1st column, formatting/spacing different for last cells (space in front)

Table 3, fonts and size of text differ significantly between cells... please standardize. (alternates randomly Calibri - Arial, and font size 10 - 8, for example)

Figures 1 and 2: Please ensure the "boxes" of the graphs are aligned. Please see my attachment where I outline some alignment issues. Nothing major but will improve look and feel of figures.

Please check the fonts of all captions for figures and tables - some are Arial, some are Roboto and some are Times New Roman.

Thank you!

Annotated reviews are not available for download in order to protect the identity of reviewers who chose to remain anonymous.

---

## Round 0.3 · accepted · Accept

Thank you for further revising your work, and addressing all the concerns raised. The authors have benefited greatly from the peer review process, which greatly improved the quality of their work.

The revised manuscript can now be accepted for publication. Thank you authors for finding PeerJ as your journal of choice, and looking forward to your future scholarly contributions.

Congratulations and very best regards.